# A Pharmacist’s Role in a Case of Allergy Labeling and Acute Bacterial Rhinosinusitis Treatment

**DOI:** 10.3390/pharmacy12010008

**Published:** 2024-01-03

**Authors:** Megan R. Undeberg, Dana R. Bowers, Cindy N. Chau, Kimberly C. McKeirnan

**Affiliations:** Pharmacotherapy Department, College of Pharmacy and Pharmaceutical Sciences, Washington State University, Spokane, WA 99202, USA; dana.bowers@wsu.edu (D.R.B.); cindy.n.chau@wsu.edu (C.N.C.); kimberly.mckeirnan@wsu.edu (K.C.M.)

**Keywords:** rhinosinusitis, pharmacist intervention, penicillin allergy, allergy labeling, medication safety

## Abstract

This case report describes a pharmacist’s intervention with a 58-year-old female who presented with recurrent rhinosinusitis symptoms and limited treatment options due to a complicated allergy history. Using guidelines for treatment of acute bacterial rhinosinusitis coupled with a thorough antibiotic allergy assessment, the pharmacist developed a treatment plan that was acceptable to both the patient and the provider. Pharmacists can play an essential role in verification of allergies to both medications and non-pharmaceutical products, which further ensures patient safety as well as optimization of appropriate treatment methods.

## 1. Introduction

Rhinosinusitis (RS) is inflammation of the sinus and nasal cavities. Adult RS includes the following symptoms: nasal blockage or discharge plus either facial pain/pressure or reduction in or loss of smell [1]. Adult rhinosinusitis can be classified into four groups, based on the duration of symptoms: acute (symptoms lasting less than four weeks), subacute (symptoms lasting between four and twelve weeks), chronic (symptoms lasting over twelve weeks), and recurrent (four episodes lasting less than four weeks but with complete resolution between each episode occurring within one year) [2]. It is important to distinguish between the various types of rhinosinusitis as treatments vary.

Approximately 90–98% of acute RS is caused by viral infection [3]. Common viral causes for RS are similar to the viral causes of upper respiratory tract infections (URI): rhinovirus, influenza and parainfluenza [1]. Acute viral RS can transition to acute bacterial rhinosinusitis (ABRS) when bacterial growth occurs in obstructed sinus passages. While bacterial causes of RS are not as prevalent (approximately 2–10% of cases), common bacteria isolated from patients with ABRS are *Streptococcus pneumoniae*, *Haemophilus influenzae* and *Moraxella catarrhalis* [3,4]. The distinction between viral and bacterial etiologies is an important one to make when considering antibiotic selection and duration. 

Viral etiologies are presumed if patients present with symptoms for less than 10 days and have not worsened during that timeframe [4]. The diagnosis of ABRS is determined if the patient presents with any of the following: (1) persistent symptoms or signs compatible with acute rhinosinusitis, lasting for >10 days without any evidence of clinical improvement; (2) onset of severe symptoms or signs of high fever (>39 °C or 102 F) and purulent nasal discharge or facial pain lasting for at least three to four consecutive days at the beginning of illness; or (3) onset of worsening symptoms or signs following a typical viral URI that lasted 5–6 days and was initially improving (“double-sickening”) [4].

Antimicrobial stewardship (AMS) concerns the appropriate use of antimicrobials [5]. Pharmacists play a critical role in effective antimicrobial stewardship in a variety of settings. Pharmacists can help determine whether antibiotics will be beneficial to patients. Antibiotics are not effective for bacterial infections and therefore pharmacists should be familiar with criteria for ABRS [4]. Once the appropriateness of antibiotics has been determined, pharmacists can assess other patient-specific factors that are considered such as allergies, concurrent medications, and comorbid diseases for appropriate therapy selection. 

Pharmacists play a significant role in allergy assessment. Antibiotic allergies are commonly reported; although, few are true allergic reactions [6]. Patients with antibiotic allergies receive second-line therapies resulting in worse health outcomes [7]. This makes the critical evaluation of allergy assessment even more important. The following case highlights current issues with allergy labels and the vital role pharmacists play in AMS to ensure optimal therapeutic outcomes.

## 2. Case Presentation

A 58-year-old white Caucasian female (pronouns: she/her/hers) was referred by a clinic physician to an ambulatory care clinical pharmacist. The referral was for a shared medication review and intervention for therapeutic options for recurrent sinus infections in a patient with extensive allergies. The patient had a past medical history of a screw and plate placement in the right hand, radius, and foot related to a traumatic injury; fibromyalgia; mixed depression and anxiety; lower back pain; back spasms; chronic migraine; chronic obstructive pulmonary disease (COPD); gastroesophageal reflux disease (GERD); Barrett’s esophagitis; hyperlipidemia; insomnia; seasonal allergic rhinitis; deep vein thrombosis (DVT) with pulmonary embolism (PE); and insomnia. Her list of allergies and intolerances to medications was quite extensive as shown in Table 1. At the time of the pharmacist visit, she was smoking half a pack of cigarettes per day but expressed interest in smoking cessation. The patient’s medication list is documented in Table 2.

Prior to her consultation with the pharmacist, the patient experienced post-nasal drip, congestion, a dry cough, and a sore throat. She met with her physician and was initially diagnosed with a viral URI. She was treated with supportive care including rest, hydration, and acetaminophen for any discomfort, and was instructed to increase her ipratropium/albuterol aerosolized nebulizer treatments from twice daily to four times daily. 

During her visit with the pharmacist three weeks later, the patient was struggling with bilateral maxillary swelling, facial tenderness, pressure on the left side of the forehead and cheek, nasal congestion, and a dry cough. She did not have dyspnea or a change in smell or taste. The patient stated these symptoms had been ongoing for several weeks; her symptoms had improved initially after she saw her physician, but now seemed to be worse than before. Since her physician had recommended supportive care, she had started taking guaifenesin 600 mg by mouth twice daily, saline nasal spray, neti pot rinses with saline, and Dayquil (acetaminophen, dextromethorphan, and phenylephrine) and Nyquil (acetaminophen, dextromethorphan, and doxylamine) over-the-counter preparations. None of these self-treatment methods had brought her any relief.

During the consultation, the pharmacist identified a possible case of ABRS. While the patient was present, the pharmacist consulted with the physician, and discussed options to manage rhinosinusitis in a patient with an extensive allergy list. Due to the duration of presentation, patient symptoms, and a classic “double sickening” presentation, the patient was classified as having ABRS. According to the current guidelines, in addition to supportive symptom management, treatment with an antibiotic was indicated. There are two main clinical practice guidelines for ABRS: the American Academy of Otolaryngology—Head and Neck Surgery Foundation (AAO-HNSF) [4], updated in 2015, and the Infectious Diseases Society of America (IDSA) [2], updated in 2012. Table 3 summarizes the empiric antibiotic regimens.

Since this patient has a reported anaphylactic allergy to the first-line therapy (amoxicillin), the second-line therapy was considered. However, a critical evaluation of the patient’s reported allergy through a detailed allergy history prior to determining a final antimicrobial agent for treatment is important [16].

## 3. Role of the Pharmacist: Antibiotic Allergy Assessment

Patient-reported allergy to penicillin antibiotics is the most commonly reported drug allergy, with an estimated prevalence of 10% in the general population and up to 15% of hospitalized patients [16]. While reported allergies to penicillin antibiotics are common, most patients are not truly allergic to penicillin after further assessment [6]. The problem with a patient having a penicillin allergy label, accurate or not, is that these patients do not receive the preferred antibiotic therapy. Patients with a penicillin allergy label are exposed to more antibiotics, have increased risk of Clostridioides difficile, methicillin-resistant Staphylococcus aureus, vancomycin-resistant Enterococci infections, and have longer hospital stays [7].

Due to the potential impact of allergy labeling, careful allergy history, and reconciliation is an important role for healthcare providers and particularly for pharmacists. Current IDSA guidelines suggest allergy assessment and penicillin skin testing in patients with a history of beta-lactam allergy [5]. Components of a thorough medication allergy history include gathering the reported allergy medication and any other concurrent medications being taken when the reaction occurred, the associated symptoms of the reaction, the chronological relationship from reaction onset and reaction, the management of the reaction, and if any similar medications have been administered since [17].

After thoroughly conducting a patient-specific allergy history and complete medication reconciliation, the patient can be further stratified into low or moderate/high risk antibiotic allergy history categories. The patient may be considered low-risk if the details of the reaction include the following: signs and/or symptoms associated with drug intolerance (isolated gastrointestinal upset, chills, headache, or fatigue), family history, itching, unknown reaction or remote reaction (>10 years ago) [18,19]. If the patient is determined to be low risk, the next step is to perform a direct challenge either as a one-step or two-step/three-step graded challenge regimen [20]. A one-step challenge involves giving the patient a full therapeutic dose [20]. This is appropriate for patients with a family history but no personal history, drug intolerance, pruritis, or other non-immune related reactions. A two- or three-step graded challenge is appropriate for patients with risks that are slightly higher than indicated for a one-step, but still relatively low risk overall [20]. Briefly, the patient receives a small test dose and then is observed for reactions for 30–60 min before receiving a full dose of the medication [20]. This is done to rule out any immediate reactions that may occur.

Patients may be considered moderate to high risk if their allergy history and reconciliation reveal any potential IgE mediated reactions such as anaphylaxis, cough, throat tightness, shortness of breath, angioedema or swelling, or other reactions that are not considered low risk [13,14]. Penicillin skin testing is considered the standard of care. Briefly, this is a skin-prick test, followed by an intradermal test if the skin prick test is negative [21]. Negative intradermal results are then confirmed by an oral amoxicillin challenge. This type of testing can be performed in various settings by different healthcare professionals [21]. Direct challenges are recommended over skin testing in patients with non-anaphylactic, non-severe cutaneous drug allergy histories [20]. This has been recently validated in the PALACE trial. Direct oral challenge was determined to be safe and effective in low-risk patients with penicillin allergy compared to penicillin skin testing [22]. Utilizing this method can help de-label patients in a relatively faster and less labor-intensive manner. 

In this patient case, the patient reported an anaphylactic reaction to amoxicillin and further investigation was warranted. She was interviewed and completed an allergy history and reconciliation with the pharmacist. The patient reported that she was told as a child she had a “reaction” to amoxicillin, but without any further details. When asked about whether treatment was needed when she experienced the reaction to amoxicillin, the patient stated that she was told to stop taking the amoxicillin only and did not require additional interventions. She has not had any penicillin or related antibiotics since. Based on this history, the patient was determined to be at a low risk for amoxicillin allergy. She was administered a direct challenge with a full dose amoxicillin 875 mg by mouth twice daily without any immediate or delayed reactions. the patient completed a course of first-line therapy for ABRS with amoxicillin/clavulanate without recurrence of sinus congestion, maxillary and sinus pain and pressure, post-nasal drip, or dry cough.

## 4. Conclusions

Pharmacists play a critical role in allergy assessment and reconciliation. By investigating patient allergy and reaction details, pharmacists can identify instances where a true allergy is unlikely. Appropriate allergy labeling directly impacts patient care and access to safe, effective, and appropriate medication therapy. By documenting accurate allergy information, pharmacists can ensure patients receive appropriate antibiotics and optimal care.

## Figures and Tables

**Table 1 pharmacy-12-00008-t001:** Patient’s documented allergies and associated reactions.

Allergy	Reaction
Amoxicillin	Anaphylaxis
Aspirin	Hives
Codeine	GI Upset
Diclofenac-Misoprostol	Rash
Gabapentin	Swelling of legs
Hydrocodone	GI Upset
Ketorolac	GI Upset
Morphine	GI Upset, itching
Nitrofuran Derivatives	Rash
Pregabalin	Swelling of legs; edema
Sumatriptan	Rash
Paroxetine	Paradoxical anxiety
Onion	Anaphylaxis

Abbreviation used: GI: gastrointestinal.

**Table 2 pharmacy-12-00008-t002:** Patient medication list at the time of the first visit with the pharmacist.

Generic Medication and Strength	Dose	Indication(s) for Use
Albuterol HFA 90 mcg per actuation inhaler	Inhale two puffs PO every 6 h as needed for shortness of breath	COPD, shortness of breath
Ipratropium/albuterol nebulized solution 0.5 mg/3 mg per vial	Nebulize contents of one vial PO every 6 h as needed for wheezing	COPD, shortness of breath
Rivaroxaban 10 mg tablet	Take one tablet PO once daily	Stroke prevention
Atorvastatin 40 mg tablet	Take one tablet PO once daily	Hyperlipidemia
Esomeprazole 40 mg capsule	Take one capsule PO once daily	GERD
Lidocaine 5% patch	Apply one patch for 12 h, then remove patch for 12 h	Chronic back pain
Metaxalone 800 mg tablet	Take one tablet PO once daily	Chronic back pain
Fentanyl 50 mcg patch	Apply one patch every 72 h. Remove old patch when applying new.	Chronic back pain
Hydromorphone 4 mg tablet	Take one tablet by mouth every 6 h as needed for breakthrough pain	Breakthrough pain
Naloxone 4 mg nasal spray	Deliver one spray to one nostril if needed for overdose	Opioid overdose
Zolpidem 10 mg tablet	Take one tablet PO at bedtime for sleep	Insomnia
Hydroxyzine pamoate 25 mg tablet	Take one capsule up to three times daily for anxiety	Anxiety
Promethazine 25 mg tablet	Take one tablet PO every 4 h as needed for nausea	Nausea
Nicotine 21 mg patch	Apply one patch every 24 h for smoking cessation	Smoking cessation
Estradiol 0.1 mg/gm vaginal cream	Insert 1 g vaginally for discomfort and irritation	Post-menopausal vaginal atrophy
Cholecalciferol 2000 international unit capsule	Take one capsule PO once daily	Vitamin D supplementation

Abbreviations used: PO: by mouth; COPD: chronic obstructive pulmonary disease; GERD: gastroesophageal reflux disease; gm: gram; mcg: microgram; and mg: milligram.

**Table 3 pharmacy-12-00008-t003:** Antimicrobial Therapy for acute bacterial rhinosinusitis [2,4].

	IDSA Recommendations [4]	AAO-HNSF Recommendations [2]
Preferred therapy	Amoxicillin-clavulanate [8] 500 mg/125 mg by mouth three times daily 875 mg/125 mg by mouth twice daily 2000 mg/125 mg by mouth twice daily	Amoxicillin +/− clavulanate Amoxicillin monotherapy [9] 500 mg by mouth three times daily 875 mg by mouth twice daily 2000 mg by mouth twice daily Amoxicillin-clavulanate 500 mg/125 mg by mouth three times daily 875 mg/125 mg by mouth twice daily 2000 mg/125 mg by mouth twice daily
Alternate therapy—penicillin allergy	Doxycycline [10] 100 mg intravenously or by mouth twice daily 200 mg intravenously or by mouth daily	
	Levofloxacin [11] 500–750 mg intravenously or by mouth daily	
	Moxifloxacin [12] 400 mg intravenously or by mouth daily	
Non-type I hypersensitivity to penicillin	Clindamycin plus (cefixime or cefpodoxime) Clindamycin 150–300 mg by mouth four times daily [13] Cefpodoxime 200 mg by mouth two times daily [14] Cefixime 400 mg by mouth daily or 200 mg by mouth twice daily [15]	
Duration of therapy	5 to 7 days (adults) 7 to 10 days (children)	5 to 7 days

Abbreviations used: IDSA: Infection Diseases Society of America; AAO-HNSF: American Academy of Otolaryngology—Head and Neck Surgery Foundation; and mg: milligrams.

## Data Availability

Data is unavailable due to privacy or ethical restrictions.

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
