# Peer review of "A Pharmacist’s Role in a Case of Allergy Labeling and Acute Bacterial Rhinosinusitis Treatment"

_pharmacy, 2024, doi:10.3390/pharmacy12010008_

Round 1

Reviewer 1 Report

Comments and Suggestions for Authors

First off I would like to thank the Authors for this submission. Information that contributes to highlighting new evidence and works to change the culture surrounding beta lactam allergy labeling and prescribing is a necessity so that our guidelines and practices can begin to catch up to the newer practices that improve patient centered outcomes and antimicrobial stewardship practices. 

The introduction is only focused on information regarding rhinosinusitis. It does not include information regarding pharmacists role or background information highlighting the importance of appropriate allergy labeling or the current issues with penicillin allergy labeling and how it affects patient outcomes and AMS. It doesn’t set the stage well for the importance of the article and how it adds value to current literature. Would strongly suggest scaling down on background surrounding diagnosis of RS and enhancement of these other points.

Table 1 should have the headers of allergy and reaction presented different from the text of the rest of the table so it is set apart. At its’ current state it is confusing to readers.

Suggest improved formatting to enhance readability for Table 2. 

Line 123 modify this sentence so it reads appropriately: “After carefully allergy history and reconciliation, the patient can be further stratified into low or moderate/high-risk antibiotic allergy history categories.”

Highly recommend incorporating the PALACE trial into this article to enhance current relevancy. Copaescu AMVogrin SJames F, et al. Efficacy of a Clinical Decision Rule to Enable Direct Oral Challenge in Patients With Low-Risk Penicillin AllergyThe PALACE Randomized Clinical TrialJAMA Intern Med. 2023;183(9):944–952. doi:10.1001/jamainternmed.2023.2986

Comments on the Quality of English Language

Very mild correction as stated above: Line 123 modify this sentence so it reads appropriately: “After carefully allergy history and reconciliation, the patient can be further stratified into low or moderate/high-risk antibiotic allergy history categories.”

Author Response

Please see the letter attached.

Reviewer 2 Report

Comments and Suggestions for Authors

This is a well written paper that outlines a case-study and literature review for amoxicillin allergy assessment by a pharmacist. The literature review is reasonably well constructed but it is a narrative review rather than a systematic review.  The manuscript could be strengthened by a more systematic approach to the review.

Comments on the Quality of English Language

Good.

Reviewer 3 Report

Comments and Suggestions for Authors

Dear authors,

Thank you for the opportunity to review this manuscript.

The present work is interesting and relevant from a clinical point of view, because the involvement of the clinical pharmacist in this new field of screening the patient with a medical history of beta-lactam allergy, is a future challenge for this profession.

Below please find my most important comments:

 Page 1, section Introduction, row 21: „ Rhinosinusitis (RS) is an obstruction of ……” → «Rhinosinusitis (RS) is inflammation of the nasal cavity and paranasal sinuses …» (World Allergy Organization - definition); Nasal blockage/obstruction is the consequence of the sinonasal inflammatory process, not vice versa. Yes, the presence of nasal polyps is an obstructive factor favoring the development of rhinosinusitis, but it is not a mandatory condition to be present in all patients with rhinosinusitis. In this context, I suggest correcting and reformulating the definition of rhinosinusitis.

Page 1, section Introduction, row 24: „Rhinosinusitis can be classified ....” = Adult Rhinosinusitis can be classified..... It is necessary to mention that it is about the forms of rhinosinusitis in adults, because in pediatrics there is no subacute clinical form.

Page 1, section Introduction, row 26-27: „Rhinosinusitis …. recurrent (four episodes lasting less than four weeks but with complete resolution between each episode)”

Recurrent Acute Rhinosinusitis definition = Four or more episodes of ARS per year. Please mention and correct that the time interval in which the patient must manifest the four episodes of acute rhinosinusitis to be considered the recurrent clinical form of acute rhinosinusitis is 1 year.

All the comments made are in agreement with International Consensus Statement on Allergy and Rhinology (2021): Orlandi, R. R., Kingdom, T. T., Timothy L. Smith, Benjamin Bleier, ......., Bing Zhou. International Consensus Statement on Allergy and Rhinology: Rhinosinusitis 2021. Int Forum Allergy Rhinol.2021;11:213-739; DOI: 10.1002/alr.22741)

In this context, I also recommend an update of bibliographic note 1, from 2016. In 2021, a new International Consensus Statement on Allergy and Rhinology appeared.

Page 3, section 2. Case Presentation, row 86: „…. AB was classified as having ABRS” - Who is "AB"? Are the initials of the patient's name? It is not specified anywhere what this abbreviation represents. Please clarify this abbreviation.
